# Patellofemoral Arthroplasty Is an Efficient Strategy for Isolated Patellofemoral Osteoarthritis with or without Robotic-Assisted System

**DOI:** 10.3390/jpm13040625

**Published:** 2023-04-02

**Authors:** Cécile Batailler, Pit Putzeys, Franck Lacaze, Caroline Vincelot-Chainard, Andreas Fontalis, Elvire Servien, Sébastien Lustig

**Affiliations:** 1Department of Orthopaedics, Croix Rousse Hospital, Claude Bernard Lyon 1 University, 69004 Lyon, France; 2IFSTTAR, LBMC UMR_T9406, Claude Bernard Lyon 1 University, 69100 Lyon, France; 3Department of Orthopaedics, Hôpitaux Robert Schuman, L-2540 Luxembourg, Luxembourg; 4Department of Orthopaedics, ORTHOSUD, Clinique St Jean Sud de France, 34430 Saint Jean de Vedas, France; 5Department of Trauma and Orthopaedic Surgery, University College London Hospitals, London NW1 2BU, UK; 6Interuniversity Laboratory of Biology of Mobility (LIBM-EA 7424), Claude Bernard Lyon 1 University, 69003 Lyon, France

**Keywords:** patellofemoral arthroplasty, robotic arm assisted surgery, functional outcomes, revision, patellar tilt, inlay, onlay

## Abstract

There is relative paucity in the literature concerning outcomes after robotic-assisted Patellofemoral Arthroplasty (PFA). The aims were (1) to evaluate outcomes in patients undergoing PFA with inlay or onlay components, with or without robotic arm assistance and (2) to identify risk factors of poor outcomes after PFA. This retrospective study included 77 PFA for isolated patellofemoral joint osteoarthritis, assigned to three groups (18 conventional technique, 17 image-free robotic-assisted system and 42 image-based robotic-assisted system). The demographic data were comparable between the three groups. The clinical outcomes assessed were: Visual Analogue Scale, Knee Society Score, Kujala score and satisfaction rate. The radiological measures were: Caton Deschamps index, patellar tilt and frontal alignment of the trochlea. Functional outcomes, satisfaction rate and residual pain were comparable between the three groups. Patellar tilt improvement was superior when a robotic device was used (either image-based or image-free) compared to the conventional technique. There were three revisions (3.9%) at the last follow-up related to femorotibial osteoarthritis progression. Multivariate analysis found no significant risk factors for poor outcomes, with respect to the surgical technique or implant design. Functional outcomes and revisions rate after PFA were comparable between the surgical techniques and implants. Robotic-assisted systems were associated with a superior improvement of the patellar tilt compared to the conventional technique.

## 1. Introduction

Isolated patellofemoral osteoarthritis (PFOA) is common and its reported incidence in epidemiological studies is approximately 10% in patients older than 40 years [1,2] and as high as 24% in females and 11% in males in adults over the age of 55 [3]. Patellofemoral arthroplasty (PFA) is a viable and effective option, yet intermediate, for the treatment of isolated PFOA with intact ligamentous structures [4].

Trochlear components currently available in the market can be categorized into onlay and inlay. Inlay components are designed to replace the worn cartilage and mandate the creation of a bony bed within the native trochlea, following which they are implanted flush to the native cartilage [5]. Onlay philosophy is based on the trochlear cuts performed during total knee arthroplasty and components are designed to replace the entirety of the anterior trochlea [5]. Most first-generation designs were characterized by an inlay design and were associated with poor clinical outcomes and high failure rates owing to patellar maltracking, catching and anterior knee pain [5,6,7,8,9]. The evolution of implant technology had led to newer prosthesis designs, predominantly represented by onlay designs and a few second-generation inlay implants, designed to anatomically resurface the trochlea; holding promise in addressing previously reported issues and resulting in higher function levels and less mechanical complications [4,10].

The evolution of surgical technology had led to the development of robotic arm assistance for knee arthroplasty that has been reported to result in more reproducible, accurate and personalized component positioning [11,12,13,14], in addition to minimizing bone and soft tissue peri-articular trauma [15,16]. Available robotic systems encompass image-free and image-based systems that use pre- operative computerized tomography scans (CT) to create a computer-aided design model of the patient’s knee joint and guide implant positioning and bone resection. 

Despite some encouraging published results, there is paucity in the literature with respect to clinical and radiological outcomes of the newer trochlear implant designs and utility of robotic arm assistance. Thus, the aims of this study were (1) to compare clinical, radiological outcomes, and complications after patellofemoral arthroplasty with inlay or onlay implants and utilization of robotic arm assistance; (2) to identify risk factors of poor outcomes after PFA. It was hypothesized that the clinical and radiological outcomes and the complications rate were similar between the inlay and onlay implants and between mechanical and robotic-assisted techniques in the short term.

## 2. Materials and Methods

### 2.1. Study Population

This multicentric, retrospective, comparative cohort study, encompassed 77 patients undergoing PFA for isolated PFOA performed either conventionally or with robotic-arm assistance. Based on whether an imageless or image-based robotic system was utilized, the study population was assigned in 3 groups: conventional technique (Mechanical group), robotic-arm-assisted technique with an image-free system (Image-Free group) and robotic-arm-assisted technique with an image-based robotic system (Image-Based group) (Figure 1). Exclusion criteria were revision PFA or associated surgical procedures such as anterior cruciate ligament reconstruction, medial or lateral Unicompartmental Knee Arthroplasty (UKA). The minimum follow-up was one year after surgery. All procedures were performed by three high volume knee surgeons (SL, PP, FL). Demographic data are summarized in Table 1.

### 2.2. Implants

Two different implant designs were included in the study: inlay (implanted with an image-based robotic system) and onlay trochlear implant (implanted conventionally or with an image-free robotic system). Seventy-seven patients undergoing PFA across three orthopaedic departments were included: 17 onlay PFA with image-free robotic-arm assistance; 42 inlay PFA with image-based robotic arm-assistance. These groups were compared to a historical series of 18 onlay PFA performed by conventional technique. All implants in this cohort study were cemented and the same implant was used within each group. 

The implant in the Mechanical group was the Gender Solutions^®^ Patello-Femoral Joint System (Zimmer Biomet^®^, Warsaw, IN, USA). In the Image-Free group, the Journey^®^ Patellofemoral Joint System (Smith and Nephew^®^, Andover, UK), a second generation onlay component was used. This was implanted with an image free, robotic assistive device, with a bone morphing during the surgery (BlueBelt Navio robotic surgical system—Smith & Nephew^®^). In the Image-Based group, the Restoris^®^ MCK patellofemoral system was implanted utilizing the MAKO Robotic Arm Interactive Orthopedic System (RIO) based on a preoperative CT scan (Stryker Corp, Mako Surgical Corp, Ft. Lauderdale, FL, USA). 

With the Navio System, planning of implants position and bone cuts were determined intra-operatively after knee mapping, without the need for preoperative Magnetic Resonance Imaging (MRI) or Computed Tomography (CT) scan. The standard pre-operative planning included a clinical evaluation of the knee and preoperative radiographs. The patient was placed in a supine position, with one lateral and one distal positioner to hold the knee at 90°. The NAVIO setting consists of 3 elements: an infrared camera which must be installed at about 1 m from the surgical site; a touchscreen monitor covered with a sterile drape and a computer for controlling the robotic burr and irrigation during burring. The first stage was to position the femoral tracking arrays using a percutaneous drilling. Then, the approach was performed as usual. In a second time, we made the acquisitions of several anatomical points and axes. The femoral condyles and trochlea are mapped. Navio system captured during the surgery a virtual 3D model of the patient’s cartilage and bone morphology. At the end of these data acquisitions, the surgical planning can be performed. The first steps were to determine the implant size then the positioning of femoral implant in all planes of view. Arthritic cartilage and bone were then methodically removed using the burr. The system continuously tracked the position of the patient’s lower limb and the progress of bone resection. The patella resection was performed with a conventional ancillary.

With the Mako System, A preoperative CT scan was performed. The surgical planning was performed before the surgery on the scan. The implant size and the positioning of the femoral implant in all planes of view were determined based on the CT scan. The installation and the femoral tracking arrays were performed as with the Navio System. Then, the femur was matched with the CT scan with the acquisitions of several anatomical points. The planning was checked during the surgery. The surgeon can also control the patellar tracking during the flexion. If needed, the surgeon can adjust the femoral positioning to improve the patellar tracking. Finally, the robotic arm performed the bone resection with a burr according to the planning. The patella resection was performed with a conventional ancillary.

### 2.3. Data Assessment

All patients were followed according to the same protocol in the three participating centres. Weight bearing antero-posterior and lateral knee radiographs, a patellar axial view, and full-length standing radiographs, were performed pre-operatively and at the time of the last follow-up. Severity of osteoarthritis was assessed using the modified Iwano classification [17]. The following radiographic measurements were performed: hip-knee-ankle (HKA) angle, Caton-Deschamps Index [18], patellar tilt [19,20] and frontal alignment of the trochlea [21]. Caton-Deschamps index was defined as the ratio between the patellar length and tendon length. Patellar tilt was measured as the angle between the mediolateral axis of the patella and the line tangent to the anterior border of the native femoral condyles (before surgery) or between the bony cut of the prosthetic patella and the two most anterior points of the prosthetic trochlea (after surgery). The frontal alignment of the trochlea was defined as the angle between the frontal axis of the trochlear groove and the axis of the distal border of the femoral condyles. Radiological measurements were performed by independent reviewers for all measurements. Clinical outcome measures included the visual analogue scale (VAS); Knee society score (KSS) (knee and function scores); Kujala score; satisfaction rate, corresponding to the rate of subjectively satisfied or very satisfied patients. Adverse events were recorded at any timepoint and at the time of the final follow-up. All revisions, in addition to the etiology, were recorded.

### 2.4. Statistical Analysis

Continuous variables are presented using the mean and standard deviation. Categorical outcomes were compared using Fisher’s exact test and chi-squared test. Normally distributed continuous variables were compared using Student *t*-test. A *p* value < 0.05 was considered statistically significant for all analyses. 

The multinomial logistic regression model to investigate risk factors included demographic data (age, gender, BMI), surgical technique (conventional, image-free or image-based robotic systems), design of implants (inlay or onlay) and radiological measurements. For this analysis, a control group was established with patients undergoing a PFA with no post-operative complications, either satisfied or very satisfied by surgery and with no residual pain. All statistical analyses were performed with the XLSTAT™ software (AddInsoft, Paris, France).

### 2.5. Ethical Approval

All procedures were performed in accordance with the ethical standards of the institutional and national research committee, the 1964 Helsinki declaration and its later amendments, or comparable ethical standards. Data collection and analysis were carried out in accordance with MR004 Reference Methodology from the Commission Nationale de l’Informatique et des Libertés (Ref. 2226075) obtained the 19 April 2022. The study was registered and filed on the Health Data Hub website. As per institutional standards, formal patient consent is not required for this type of study.

## 3. Results

Mean follow-up was 39.6 months ± 13.3 (12–60) in the image-based robotic-assisted group versus 42.2 months ± 15.3 (12–60) in the image-free robotic-assisted group. The mechanical group comprised a historical series with a mean follow-up of 68.5 months ± 13 (46–92).

Functional outcomes (KSS scores, Kujala score), satisfaction rate and residual pain were comparable at the time of last follow up between the three groups and by comparing the two implant designs (Table 2 and Table 3). There was a superior improvement of the patellar tilt when a robotic device was used (either image-based or image-free) compared to the conventional technique (*p* < 0.0001, Table 3). The patellar tilt was significantly improved in the whole cohort after the surgery (*p* < 0.0001). No statistically significant differences in radiological measurements were noted between the image-based and image-free groups or between the inlay or onlay implants (Table 2 and Table 3).

There were three revisions (3.9%) at the time of last follow-up owing to progression of the disease and not malposition of implants. One medial unicompartmental knee arthroplasty in the image-based robotic group for symptomatic medial femorotibial osteoarthritis and two total knee arthroplasties in the image-free robotic group for symptomatic femorotibial osteoarthritis. These revisions were secondary to extended indications. There was no complication without revision at the last follow-up.

With respect to risk factors analysis, the control group was composed of 47 patients (61%). Multivariate analysis found no significant risk factors for poor outcomes, with respect to the surgical technique or implant design (Table 4).

## 4. Discussion

The most important finding of the current study was that similar functional outcomes and radiological measurements were observed after inlay or onlay PFA performed with or without robotic-assisted systems, except for a superior improvement of the patellar tilt with the robotic assisted systems. 

The first generation of inlay implants were associated with poor clinical outcomes and high revision rates in the mid- to long-term range, while patella maltracking was reported in up to 36% of cases [6,7,8,22]. The evolution of designs of trochlear implants has improved functional outcomes and has shown promising results in reducing maltracking and revision rates [4,10,23,24]. In this current study, there were satisfying functional outcomes with two designs of implants (second generations of inlay and onlay) without significant differences between both implants. These results were similar to those described in the literature with these new generations of designs [4,10,23,25]. A recent prospective study reported high satisfaction, with 80% (20 out of 25 patients) reporting a score of 8 and above after PFA performed with an image-based robotic assisted system [23]. Turktas et al. also described encouraging clinical and radiological results after PFA performed with an image-based robotic assisted system and only one revision to TKA (3.2%) at a mean follow up of 15.9 months [26]. Failures of PFA implants are associated with two types of post-operative complications: early complications due to patellar maltracking, including painful instability, subluxation, or dislocation, and late complications due to the spread of arthritis to the tibiofemoral joint [24,27]. Dejour et al. have demonstrated the importance of a good selection of patients to obtain good functional outcomes with a low rate of revisions [4]. Indeed, the only three revisions of this cohort were due to progressive femorotibial osteoarthritis, and probably due to improper patient selection. This good patient selection appears more crucial than the used designs (new generation of inlay or onlay). 

The second essential parameter for a successful PFA is the excellent positioning of implants to obtain satisfying patellar tracking and good restoration of the anterior compartment [24,28]. In this study, there was no patellar maltracking or revision for malpositioning or instability. Two reasons can explain this: the surgeon’s experience and the use of a robotic-assisted system. Indeed, PFA needs a very accurate positioning. The three surgeons were experienced with a high volume of knee arthroplasty, including PFA. The historic group with the conventional technique was the best witness of the surgeon experience because the functional and radiological outcomes were similar to the two groups with robotic-assisted systems. The robotic assisted systems used in the two other groups allowed a 3D assessment of the trochlea with a CT scan or a bone morphing. The position of the Restoris MCK or the Journey implant can be individualized in all three planes with the utilization of the robotic system based on anatomical considerations and the severity of the disease. With the image-based robotic-assisted system, the patellar tracking can also be visualized and checked during the surgery before and after the PFA implementation. A recent prospective study reported an accurate execution of the pre-operative plan with the image-based robotic-assisted system [23]. These systems have also proven their accuracy and satisfying clinical outcomes with unicompartmental knee arthroplasty [11,12,13,14,29].

The improvement of the patellar tilt was superior when robotic arm assistance was utilized. Patellar tilt is involved in the pathogenesis of maltracking and anterior knee pain [30] and, despite its multifactorial nature, is a sensitive marker for patellar instability [31]. With a conventional technique for onlay implants, the anterior femoral cut should be perpendicular to the Whiteside’s line. In patients with PFOA secondary to trochlear dysplasia, the anterior cut is recommended with a slight external rotation to accommodate the tight lateral retinaculum. Conceivably, the lack of precision with conventional instrumentation in this step could partially explain the superior improvement in the patellar tilt when a robotic system was used. Selvaratnam et al. reported precise and reproducible execution of the three-dimensional pre-operative plan when an image-based robotic system was used [23]. They reported an average difference with the final intraoperative trochlear implant position of 0.43° for rotation (r = 0.93). 

This study had several limits. First, its retrospective design may have led to the introduction of confounders and selection bias. Nevertheless, the outcomes at the last follow-up were prospective. Second, this study reflects the performance across 3 different centres, and patients were not matched. However, all operating surgeons were high volume Consultant Orthopaedic Surgeons, and a standardized follow-up protocol was utilized. Additionally, the three groups were comparable. Then, the number of patients in each group was low, with a risk of low study power. However, this surgery is uncommon. Additionally, this study is the first comparative study on robotic-assisted PFP. Lastly, the positioning assessment was performed on radiographs and not on a CT scan. The accuracy of these measurements is probably lower than with a CT scan. Nevertheless, routine CT scan is not recommended after PFA, as this would expose patients to unnecessary radiation and would not reflect common practice.

The main strength of this study was that it was the first comparative study between second-generation PFA implants evaluating outcomes with or without robotic-assisted systems. Indeed, very few studies assess robotic-assisted PFA. A clinical assessment of this progressive practice was necessary.

## 5. Conclusions

PFA was an effective strategy in patients with isolated patellofemoral osteoarthritis, resulting in satisfying functional outcomes and an improvement of the patellar tilt. Functional outcomes were comparable between conventional technique and robotic-assisted systems. No significant differences were evident between inlay and onlay implants. Robotic-assisted systems were associated with a superior improvement of the patellar tilt compared to the conventional technique.

## Figures and Tables

**Figure 1 jpm-13-00625-f001:**
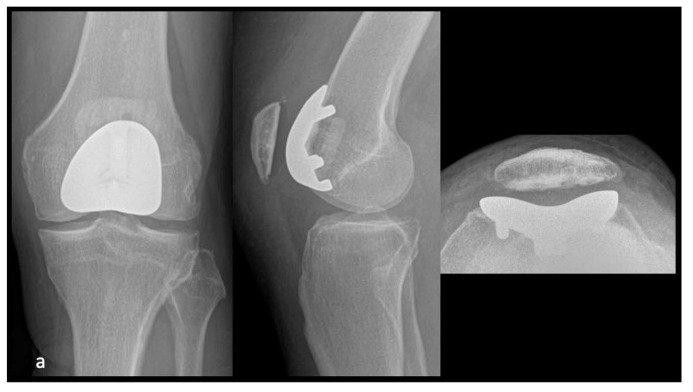
Postoperative radiographs of PFA performed conventionally (onlay implant) (**a**), with an image-free robotic system (onlay implant) (**b**) and with an image-based robotic system (inlay implant) (**c**).

**Table 1 jpm-13-00625-t001:** Demographic data and preoperative radiological parameters.

	Whole Cohort N = 77 Knees	Inlay PFA N = 42	Onlay PFA N = 35	*p*-Value	Mechanical N = 18	Image Free N = 17	Image Based N = 42	*p*-Value
Age (years)	61.4 ± 12.2	62.7 ± 11.4	59.7 ± 13	0.29	61.0 ± 13.7	58.4 ± 12.4	62.7 ± 11.4	0.21
mean ± SD [Min; Max]	[24; 88]	[45; 88]	[24; 83]		[24; 83]	[34; 78]	[45; 88]	
BMI (kg/m^2^)	27.5 ± 5.5	28.7 ± 5.7	26.0 ± 5.0	0.032 *	25.9 ± 5.6	26.1 ± 4.5	28.7 ± 5.7	0.097
mean ± SD [Min; Max]	[18; 45]	[19; 45]	[18; 38]		[18; 36]	[19; 38]	[19; 45]	
Gender (Female) (%)	60 (78%)	32 (76%)	28 (80%)	0.79	12 (67%)	16 (94%)	32 (76%)	0.12
Caton Deschamps Index	1.0 ± 0.15	1.0 ± 0.14	1.0 ± 0.17	0.94	0.9 ± 0.17	1.1 ± 0.14	1.0 ± 0.14	0.073
mean ± SD [Min; Max]	[0.63; 1.4]	[0.67; 1.4]	[0.63; 1.3]		[0.63; 1.3]	[0.8; 1.3]	[0.67; 1.4]	
Patellar tilt (°)	8.9 ± 7.0	9.2 ± 7.0	8.5 ± 7.1	0.78	5.2 ± 7.1	11.6 ± 5.8	9.2 ± 7.0	0.055
mean ± SD [Min; Max]	[−16; 28]	[−4; 28]	[−16; 24]		[−16; 13]	[2; 24]	[−3.6; 28]	
HKA angle (°)	180.9 ± 3.6	180.6 ± 3.4	181.3 ± 3.8	0.46	179.2 ± 3.0	182.8 ± 3.7	180.6 ± 3.4	0.021 *
mean ± SD [Min; Max]	[173; 191]	[175; 189]	[173; 191]		[173; 185]	[177; 191]	[175; 189]	
TT−TG (mm)	11.5 ± 5.5	10.8 ± 5.2	13.8 ± 6.1	0.092	13.8 ± 6.1	NR	10.8 ± 5.2	0.092
mean ± SD [Min; Max]	[2; 23]	[2; 22]	[5; 23]		[5; 23]		[2; 22]	
Osteoarthritis stage (Iwano)								
1	8	2	6	0.107	2	4	2	0.14
2	12	7	5		2	3	7	
3	34	17	17		7	10	17	
4	17	12	5		5	0	12	

HKA: Hip-Knee-Ankle, SD: Standard Deviation, BMI: Body Mass Index, TT-TG: Tibial Tuberosity-Trochlear Groove; NR: not reported; *: significant *p*-value (<0.05).

**Table 2 jpm-13-00625-t002:** Clinical and radiological outcomes at the last follow-up in the inlay and onlay groups.

	Whole Cohort N = 77 Knees	Inlay PFA N = 42	Onlay PFA N = 35	*p*-Value
KSS Knee score	87.8 ± 13.2	86.8 ± 10.8	88.8 ± 15.2	0.55
mean ± SD [Min; Max]	[30; 100]	[62; 100]	[30; 100]	
KSS Function score	83.7 ± 16.8	84.0 ± 14.9	83.5 ± 18.5	0.90
mean ± SD [Min; Max]	[20; 100]	[33; 100]	[20; 100]	
Kujala score	85.5 ± 17.9	88.0 ± 15	83.2 ± 20.1	0.28
mean ± SD [Min; Max]	[0; 100]	[37; 100]	[0; 100]	
Satisfaction	57.6%	48.4%	65.7%	0.24
(Very satisfied or satisfied)				
VAS	1.7 ± 1.7	1.8 ± 1.7	1.5 ± 1.7	0.45
mean ± SD [Min; Max]	[0; 7]	[0; 7]	[0; 6]	
Caton Deschamps Index	0.9 ± 0.19	0.94 ± 0.18	0.96 ± 0.2	0.65
mean ± SD [Min; Max]	[0.47; 1.78]	[0.57; 1.4]	[0.47; 1.78]	
Patellar tilt (°)	3.1 ± 4.1	2.7 ± 4.4	3.7 ± 3.9	0.32
mean ± SD [Min; Max]	[−8.4; 14]	[−8.4; 12]	[−2.2; 14]	
Improvement of Patellar tilt (°)	5.2 ± 7.0	5.6 ± 6.1	4.8 ± 8.0	0.64
mean ± SD [Min; Max]	[−17.4; 19.8]	[−3.6; 19.8]	[−17.4; 19]	
Frontal alignment of trochlea (°)	90.8 ± 3.6	90.1 ± 2.4	91.5 ± 4.4	0.11
mean ± SD [Min; Max]	[82; 103]	[86; 96.7]	[82; 103]	

VAS: Visual Analogue Scale, KSS: Knee Society Score.

**Table 3 jpm-13-00625-t003:** Clinical and radiological outcomes at the last follow-up between the three groups.

	Whole Cohort N = 77 Knees	Mechanical N = 18	Image Free N = 17	Image Based N = 42	*p*-Value Mechanical vs. Image Free	*p*-Value Mechanical vs. Image Based	*p*-Value Image Free vs. Image Based	*p*-Value Global
KKS Knee score	87.8 ± 13.2	89.3 ± 11.8	88.2 ± 18.5	86.8 ± 10.8	0.97	0.81	0.93	0.82
mean ± SD [Min; Max]	[30; 100]	[55; 100]	[30; 100]	[62; 100]				
KSS Function score	83.7 ± 16.8	86.4 ± 19.6	80.4 ± 17.2	84 ± 14.9	0.54	0.88	0.76	0.57
mean ± SD [Min; Max]	[20; 100]	[20; 100]	[36; 95]	[33; 100]				
Kujala score	85.5 ± 17.9	83.5 ± 24.2	82.8 ± 15.3	88.0 ± 15	0.99	0.67	0.61	0.55
mean ± SD [Min; Max]	[0; 100]	[0; 100]	[31; 98]	[37; 100]				
Satisfaction	57.6%	61%	70.6%	48.4%	NR	NR	NR	0.35
(% of satisfied)								
VAS	1.7 ± 1.7	1.1 ± 1.5	1.9 ± 1.8	1.8 ± 1.7	0.33	0.33	0.98	0.27
mean ± SD [Min; Max]	[0; 7]	[0; 6]	[0; 6]	[0; 7]				
Caton Deschamps Index	0.9 ± 0.19	0.96 ± 0.26	0.96 ± 0.12	0.94 ± 0.18	1	0.94	0.92	0.90
mean ± SD [Min; Max]	[0.47; 1.78]	[0.47; 1.78]	[0.7; 1.2]	[0.57; 1.4]				
Patellar tilt (°)	3.1 ± 4.1	4.5 ± 4.8	3 ± 2.8	2.7 ± 4.4	0.53	0.35	0.98	0.37
mean ± SD [Min; Max]	[−8.4; 14]	[−2.2; 14]	[0; 10]	[−8.4; 12]				
Improvement of Patellar tilt (°)	5.2 ± 7.0	−0.09 ± 7.7	8.2 ± 6.5	5.6 ± 6.1	0.004 *	0.033 *	0.4	0.87
mean ± SD [Min; Max]	[−17.4; 19.8]	[−17.4; 11.5]	[−6; 19]	[−3.6; 19.8]				
Frontal alignment of trochlea (°)	90.8 ± 3.6	91.3 ± 2.5	91.7 ± 5.8	90.1 ± 2.4	0.94	0.52	0.29	0.26
mean ± SD [Min; Max]	[82; 103]	[87; 96]	[82; 103]	[86; 96.7]				

VAS: Visual Analogue Scale, KSS: Knee Society Score; *: significant *p*-value (<0.05).

**Table 4 jpm-13-00625-t004:** Risk factors for poor outcomes after PFA: multivariate analysis.

	Odds Ratio	*p*-Value
Gender (Female)	0.79 [0.09–7.2]	0.84
Age	1.0 [0.94–1.1]	0.97
BMI	1.1 [0.99–1.3]	0.062
Patellar Tilt	0.93 [0.82–1.0]	0.22
HKA	0.99 [0.78–1.3]	0.93
Surgical Technique (Conventional)	1.8 [0.17–20.4]	0.62
Implants (Inlay)	1.0 [0.99–1.0]	1

HKA: Hip-Knee-Ankle, BMI: Body Mass Index.

## Data Availability

Data is unavailable due to ethical restrictions.

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
