# Peer review of "Patellofemoral Arthroplasty Is an Efficient Strategy for Isolated Patellofemoral Osteoarthritis with or without Robotic-Assisted System"

_jpm, 2023, doi:10.3390/jpm13040625_

Round 1

Reviewer 1 Report

This study aimed to compare clinical and radiologic outcomes and complications after patellofemoral arthroplasty (PFA) using inlay or onlay implants and robotic-assisted techniques.

This is a very interesting study, but there are some problems that should be further addressed.

1. Please explain how the study size was arrived at.

        2. The study was divided into three groups (18 conventional technique, 17 image-free robotic-assisted system and 42 image-based robotic-assisted system). Bias may occur due to the uneven sample sizes of the included studies. Therefore, limitations should be mentioned and caution should be exercised when interpreting the results.

Author Response

Thank you for your comment.

You are right. The design of the study is not perfect. 

The number of patients in each group was low, with a risk of bias. But this surgery is uncommon. And this study is the first comparative study on robotic-assisted PFP. 

We have added this limit in the manuscript.

Reviewer 2 Report

Thank you for the opportunity to review the manuscript. The review aims to assess the quality and ensure the manuscript's reliability, completeness, and consistency. It is a way to improve the manuscript, but no improvements are necessary (except for the full stop at the end of the title, which you should remove). 

Methodologically, nothing to criticise. The retrospective nature can indeed be seen as more limited in evidence. However, you point this out as a limitation, and a methodologically correctly designed retrospective cohort analysis has more value than a prospective analysis with errors in its design.

I congratulate you on the manuscript and recommend its publication.

Author Response

Thank you for your comment.

No change